# Urban Pandemic Vulnerability and COVID-19: A New Framework to Assess the Impacts of Global Pandemics in the Metropolitan Region of Amsterdam

**Yaqi Wang** [1] , **Rodrigo Viseu Cardoso** [2,*] and **Claudiu Forgaci** [2]

1   Arcplus, Shanghai 200041, China; yq.wang.seu@gmail.com
2   Faculty of Architecture and the Built Environment, Delft University of Technology,
    2628 BL Delft, The Netherlands; c.forgaci@tudelft.nl
*   Correspondence: r.o.v.cardoso@tudelft.nl

**Abstract:** This paper presents the concept of urban pandemic vulnerability as a crucial framework for understanding how COVID-19 affects cities and how they react to pandemics. We adapted existing social and environmental urban vulnerability frameworks to assess pandemic impacts and responses, identifying the appropriate components and spatial, environmental and socio-demographic variables of interest. Pandemic vulnerability depends on exposure, sensitivity and adaptive capacity features, which occur in different combinations in different parts of a city. The model was applied to the Metropolitan Region of Amsterdam (MRA) to create a map of pandemic vulnerability. This map differentiates between affected areas according to the types of vulnerability they experience, and it accurately identified the most vulnerable areas in line with real-world data. The findings contribute to clarifying the challenges brought by COVID-19, identifying vulnerability thresholds and guiding planning towards pandemic resilience.

**Keywords:** COVID-19; Metropolitan Region Amsterdam; urban vulnerability; pandemic vulnerability; urban indicators

## 1. Introduction

As cornerstones of global business networks, tourism and supply chains, cities were vulnerable to the COVID-19 pandemic. Since a cluster of pneumonia cases appeared in 2019 in Wuhan, China, COVID-19 has mutated into a global crisis that has ravaged many countries and cities in a short time. As of February 2022, there have been more than 404 million confirmed cases in 220 countries and territories, resulting in more than 5.7 million deaths [1] The coronavirus raises several social, economic and environmental issues. Research confirms that it severely affects cities [2]. Metropolitan regions in particular are places of convergence and mediation for global networks, local government bodies and mobile people, are and the fastest-growing type of human settlement [3]. We are facing an invisible enemy responsible for devastating disruptions [4].

Since the main route of spread of the disease is direct person-to-person transmission, which is partly enabled by the population concentration and mobility typical of the metropolitan model [5], the clustering of economic activities and networks in cities was one of the key drivers for the spread of COVID-19. Therefore, metropolitan areas have become fertile grounds for pandemics. The epicenters of the most severe outbreaks in the first months of the pandemic were the wealthiest metropolises of industrialized countries [6]. Moreover, the rapid global spread of the pandemic intensified the fears of disturbances to economic globalization processes, in which metropolitan regions are key players. The physical and functional global network is fragile, leading to uncertainty about the future, which fertile ground for protectionist and nationalist ideas [7].

COVID-19 is not the first widespread infectious disease affecting metropolitan areas. In recent times, there was SARS in 2003 from China [8], the 2009 H1N1 variant in North America [9], and the 2015 Zika virus from Brazil [10], which killed thousands of people, causing various short-term and long-term impacts. A growing body of research has been published over the past few years on the consequences of infectious diseases in order to learn lessons from such crises [11]. However, due to its unprecedented speed, global spread and geographical distribution, COVID-19 focused our attention on the fragilities of metropolitan regions, both resourceful frontline actors in facing the pandemic and the first to suffer from and adapt to its fallout [2].

Indeed, the initial stages of the pandemic shed light on the dynamics in metropolitan regions that trigger a heightened vulnerability to the spread and impact of the disease [12]. Therefore, an urgent question is what makes urban spaces especially vulnerable to pandemics, and we argue that the concept of *urban vulnerability* [13] is crucial to understanding and advancing these issues. Vulnerability is generally defined as the "exposure to contingencies and stress, and difficulties coping with them" [14]. Cities, especially large metropolitan areas, are related to poverty and underdevelopment [15], which impact exposure to risks of infection and access to treatment. However, vulnerability is also a product of socioeconomic shocks, due to which the coping ability of marginalized groups diminishes as the difficulty of managing and prioritizing an increasingly complex set of assets increases [16], namely, in the fast-changing conditions of cities experiencing public health crises. Furthermore, during global pandemics, non-vulnerable groups can turn vulnerable due to inappropriate policy responses [17], making risks transverse the whole society.

Various negative drivers, such as population growth, climate change impacts and pollution, may increase the frequency and severity of further pandemics [18]. At the same time, future metropolitan regions will still be globally connected by all kinds of networks. The cities and towns within these regions remain interdependent, both spatially and functionally. Therefore, it is necessary to better explore the underlying factors and dynamics of pandemic vulnerability in the context of globalization and urbanization to develop the capacity of cities to react through adaptation measures [19]. It is urgent to construct and test different frameworks of *urban pandemic vulnerability* which are able to measure and differentiate types of vulnerability and guide innovative responses. However, most current definitions of urban vulnerability focus on social and environmental aspects. There has been little research explicitly exploring the definition and concept of pandemic vulnerability, although in the last year increasing attention has been paid to the reasons why cities are vulnerable to the pandemic for better prediction and preparedness [20].

We aimed foremost to propose a potential definition and framework for pandemic vulnerability, building upon existing concepts of urban vulnerability. If successful, such a framework can be regarded as a tool for policymakers to differentiate and clarify the main problems for different groups and spaces and propose mitigation strategies. Forecasting the directions and the mechanisms of impact of COVID-19 at the current stage is complicated due to incomplete research and a number of possible scenarios of spread. At the time of writing, COVID-19 has not yet been fully controlled worldwide. However, it is urgent to clarify the various challenges that cities are facing and allow more specific response strategies to be put forward according to different problems.

This study begins with an overview of the literature on the drivers of urban vulnerability, particularly climate and social crises. We discuss their points of contact with the emerging literature on the impact of COVID-19 in cities and ask to what extent elements of climate and social vulnerability can be adapted to construct a definition and indicators for a pandemic vulnerability framework. We then examine the framework in a real-life setting, the Metropolitan Region of Amsterdam (MRA), one of the regions in the Netherlands most seriously affected by the pandemic. We applied the model to assess the most vulnerable areas in the MRA, testing the compatibility between the framework and the actual number of coronavirus cases over time. We argue that according to the assessment framework, the MRA does not meet the thresholds for a vulnerable area in various ways. However,

these weaknesses can also be considered opportunities, brought by necessity, to consider alternative actions towards resilient and sustainable urban development, either by staying below or moving the thresholds of vulnerability.

## 2. A New Interpretation of Urban Vulnerability

### 2.1. Definitions of Urban Vulnerability

There are many different definitions of urban vulnerability. In general, vulnerability depends on the behavior of urban areas in face of a threat [21]. Most current discussions on the topic focus on two dimensions: social vulnerability, including poverty, insecurity or resource depletion; and environmental (or climate) vulnerability to heat waves, flooding, wildfires, etc. Social vulnerability can be assessed by the ability to deal with hazards from a socioeconomic perspective [15]. It can refer to the instability in well-being of individuals or communities in the face of changes in the form of "sudden shocks, long term trends or seasonal cycles" [16]. Numerous studies focusing on social vulnerability highlight that cities are the most vulnerable areas and suffer the worst impacts from the negative externalities of the concentration of population, economy, specialization and innovation [22]. Such externalities culminate in a limited ability to protect residents from certain socioeconomic risks and their negative consequences, including unemployment, poverty, social inequality and decreasing purchasing power (Figure 1).

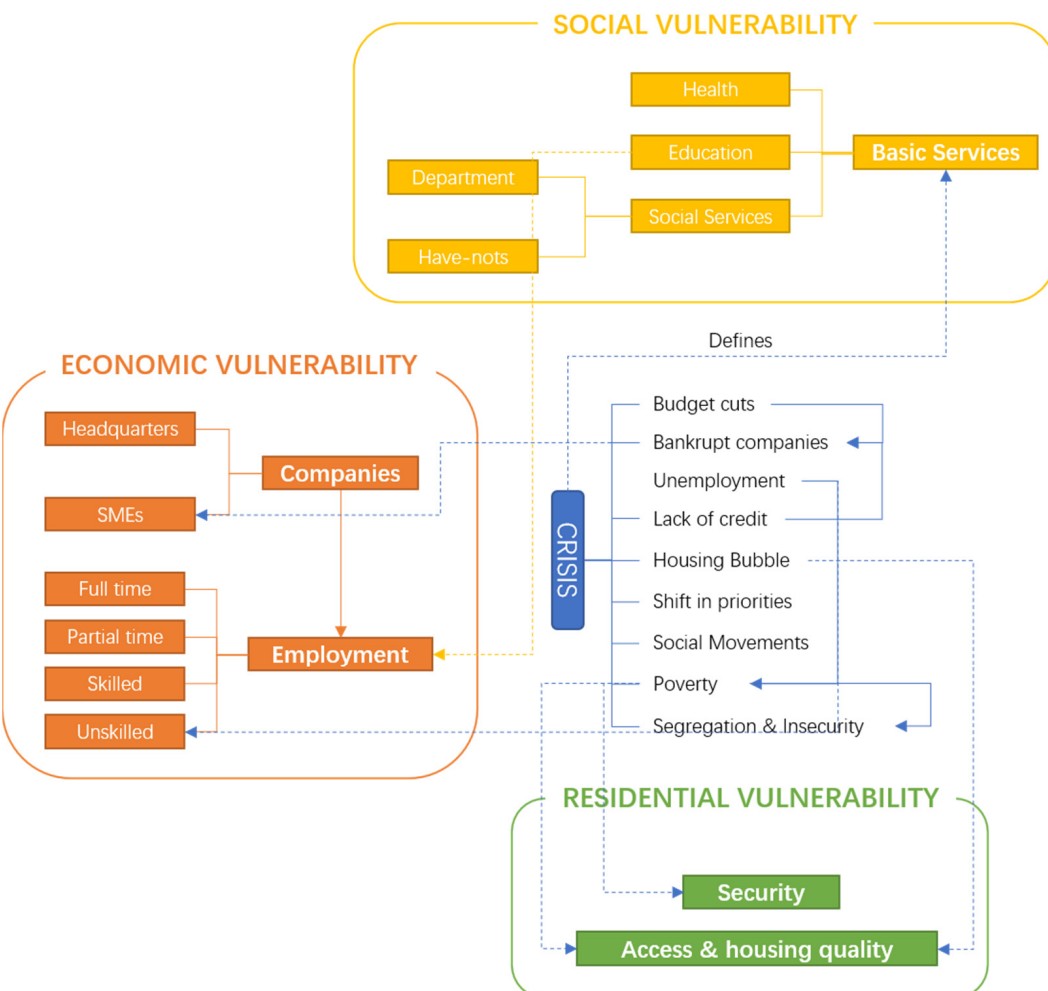

**Figure 1.** A conceptual framework of economic, social and residential vulnerability. Scheme by authors, adapted from Piñeira Mantiñán & Trillo-Santamaría, 2016).

Climate vulnerability relates to the likelihood that urban systems will be negatively affected by hazards or disasters [23]. Even though there are different definitions with many qualifiers [24,25], climate vulnerability is often represented as being impacted by both internal properties and external drivers of three key types: exposure, sensitivity and adaptive capacity (Figure 2). Exposure involves likelihood: it defines how fragile the environmental conditions of a system become to a potential threat, for instance, through the accumulated effects of detrimental activities. Sensitivity entails impact: it measures the extent to which a system is affected by the actual events. Adaptive capacity means the amount, accessibility, and degree of control of livelihood assets that the system can exert in response [13]. Several experts researching vulnerability have also emphasized the need to use the term only in specific situations. Brooks writes that one can only talk meaningfully about the vulnerability of "a specified system to a specified hazard or range of hazards" [26] and differentiates between current and future vulnerability.

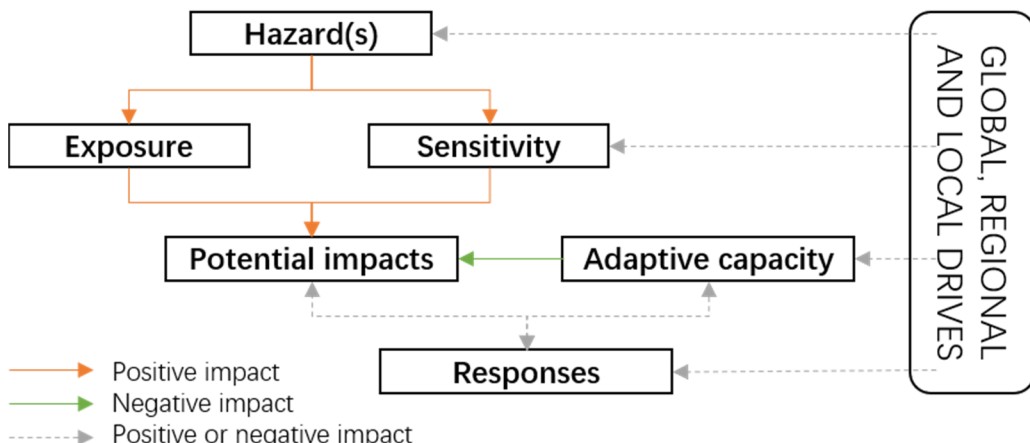

**Figure 2.** A conceptual framework of urban vulnerability to climate change. Scheme by authors, adapted from Romero-Lankao & Qin, 2011.

*2.2. Pandemics as a Socio-Environmental Event: Constructing an Urban Pandemic Vulnerability Framework*

The concept of urban pandemic vulnerability developed here is based on these references and describes threat posed by contagious diseases and their impacts on public health and socioeconomic conditions in urban areas. The reason for this convergence is that vulnerability to a pandemic is a socio-environmental issue, which can be seen as including elements of climate vulnerability and social vulnerability. On one hand, the occurrence and response mechanisms of pandemic vulnerability are similar to those of climate vulnerability, as both are caused by spatially expansive external factors (viruses and natural hazards, respectively) and ultimately respond to them at different spatial scales. On the other hand, pandemic vulnerability has a similar impact on urban systems to *social vulnerability*, including effects on the economy, residents' well-being and institutions. Finally, both the impacts and the responses can only be meaningfully assessed in a specific context and hazard. By combining these aspects, a new definition of urban pandemic vulnerability was created:

> *Urban pandemic vulnerability is the extent to which an urban system is susceptible to pandemic occurrences, as measured by the impacts to its urban spaces, social groups, and institutions.*

Accordingly, the proposed conceptual framework (Figure 3) can be explained as follows:

- Socio-environmental *external drivers* that are generally accepted in the literature are proposed as the first level of the framework, either as intensifiers or relievers: environmental conditions, the processes of economic globalization and the socioeconomic features of the population.

- *Pandemic impacts* are then defined as outcomes manifested by shocks to urban spaces (restrictions in the uses of space, functional reconversions, decay, etc.), social groups (health disruption, economic impacts, psychological stress, social conflict etc.) and institutions (institutional failures, contestation, policy effectiveness, etc.). Some of these components are also relevant to regional economic impacts of pandemics [27]
- Then, three assessment components are used to construct a vulnerability measurement, all of which can be influenced by the external drivers: *exposure* means the type and amount of pandemic risk the system is subject to. *Sensitivity* is the extent to which the system is affected by shocks that it cannot absorb. *Adaptive capacity* is the ability of the system to adjust to a pandemic and cope with its consequences.
- Therefore, the optimal *policy response* for urban pandemic vulnerability can be expressed as decreasing exposure, decreasing sensitivity, and increasing adaptive capacity through short-term and long-term measures.
- Finally, the *actionability* of the three components varies: policymakers can take quicker action to increase adaptive capacity through changes in processes and policies, than to deal with the main determinants of exposure, for example, which are harder and slower to transform.

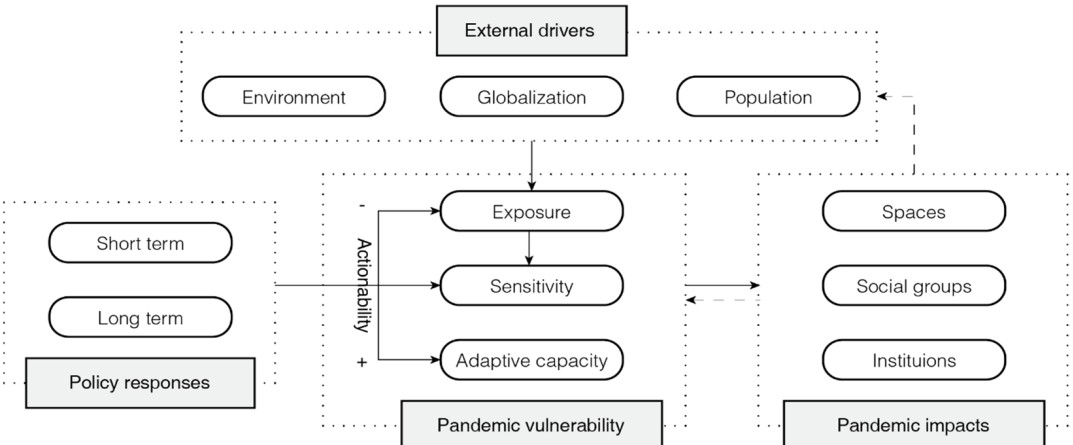

**Figure 3.** The conceptual framework of urban pandemic vulnerability.

### 2.3. Indicators of Pandemic Vulnerability

Each of the three framework components—exposure, sensitivity and adaptive capacity—should include different indicators to assess pandemic vulnerability. There are several possible applications of indicators, including identification of mitigation targets, vulnerable groups, regions and sectors; raising awareness; policy guidance; and scientific research [28]. The analysis of vulnerability is more useful when spatial characteristics are taken into account. Füssel [24] argues that identifying a vulnerable situation often involves a spatial reference and its specific attributes in relation to a specific hazard. Therefore, indicators of pandemic vulnerability in this study are a set of place-specific analytical elements informed by a spatial perspective in the context of COVID-19.

A vulnerability index with some similar features was recently presented by SecDev, a Canadian consultancy group [29]. Their approach is based on the US Center for Disease Control social vulnerability index and ranks a series of indicators to create a detailed map of pandemic vulnerability (https://urbanresilience.secdev.com/amsterdam/info, (accessed on 17 November 2021)). However, while there are several overlaps between the two approaches—including Amsterdam being used as an example—the indicators used by SecDev are population-based (economic, demographic, health, etc.) and exclude the environmental and spatial dimensions covered here. We therefore argue that, while acknowledging the value of other approaches, the framework presented here is more

complete, and by attaching vulnerability assessments to urban space features, may also be of greater interest to urban research and policy.

As a result, and in addition to population-based indicators, our framework adapts the structure of the European Environment Agency (EEA) indicators based on definitions by the Intergovernmental Panel on Climate Change (IPCC) about urban ecological vulnerability to climate-related risks, types of determining factors and responses [30]. To construct the final table, first we reviewed literature on the implications of COVID-19 to cities, planning and urban design, in order to extract relevant indicators appearing regularly as factors of urban vulnerability. Articles unrelated to those factors were excluded, such as those with a pathological or psychosocial lens, or about potentially positive pandemic impacts, such as improving air quality and decreasing congestion. After examining the selected literature, the categorization was refined according to the three components of the framework, their subcomponents and the main areas of pandemic impact, as explained above. Table 1 gives an overview of the key indicators, which are divided into short and long term according to their stability in time and capacity for change.

**Table 1.** System of pandemic vulnerability factors in different stages; ▢ short term, ▨ long term.

| Component | Exposure | | | Sensitivity | | | Adaptive Capacity | |
|---|---|---|---|---|---|---|---|---|
| | Environment | Flows | Social | Economic | Infrastructural | Biophysical | Awareness | Ability |
| **Space** | air pollution | import/ export nodes | residential density | | road/rail traffic density | open space | | hospital beds |
| | air dryness | | housing conditions | | hospital accessibility amenity accessibility | green space | | |
| **Social groups** | | daily air/rail/ road passengers | age distribution | household income | motorized dependency | | risk perception | insurance |
| | | | immigrants | migrant workers | | | education level | health |
| | | | discrimination of foreigners | | | | | ability to use technology |
| **Institutions** | | delivery systems/ supply chains | | tourism/ manufacturing | | | government trust | government integration |
| | | international corporations | | small businesses | | | | government efficiency |

### 2.3.1. Exposure

The exposure indicators cover two main issues: environmental factors focusing on air quality and factors influencing population and economic flows. First, there is some evidence that the spread rate of COVID-19 is related to air quality. According to an early study in Italy, it spread faster in northern Italy, which has higher air pollution levels [31], which, in turn, may weaken the respiratory health of vulnerable groups [32]. Studies have also found that drier air facilitates viral transmission, whose propagation is more likely to decrease in a humid environment [33]. As the coronavirus is an airborne virus, other environmental parameters, such as soil and water pollution, have limited impacts.

Flow intensity is also a relevant exposure factor. Lin et al. [34] identified population movement, namely, in Wuhan, China, as a major vector of the early spread. Their study suggests that governments need to implement more protection responses in areas with frequent people flows, such as shopping centers, airports and railway stations. Restricting the movement of people has been unsurprisingly recognized and implemented as one of the main ways to curb the spatial spread of the coronavirus. Another type of systemic exposure relates to the effects of interrupting economic flows. For instance, during the economic shutdowns caused by border closures, global supply chains have been strongly affected,

including food supply in many cities [35]. Napierala et al. [36] also pointed out that the post-pandemic recovery of the most internationalized firms in Poland is complicated and more dependent on the state of global trade and travel restrictions than other factors. This suggests that the more internationally connected firms cities host, the more impacts they suffer from the pandemic.

### 2.3.2. Sensitivity

Relevant sensitivity indicators are social, economic, infrastructural and biophysical sensitivity. While some indicators are relevant for both short-term and long-term impacts (e.g., households with low income have limited access to medical care in the short term and are more sensitive to financial difficulties in the long term), other indicators are specific to a certain period.

Social sensitivity includes well-known problems now exposed in a new light, including social inequality. Generationally, there is recognition of elderly people being a high-risk population regarding COVID-19 because of their weaker immunity, and often higher risk of disability, limited financial resources and social isolation. If elderly people rely on others for health care and daily living, the "homestay" measures present a significant challenge to ensure their basic needs. Ethnicity and culture may also play a role, particularly if immigrants suffer more from the impacts of the pandemic, due to limited accessibility to health care and higher sub- and unemployment rates. For instance, the Black and Latino death rate in New York was twice as high as Whites' until June 2020 [37]. Poor people living in crowded households, unfit sanitation, and precarious livelihoods, as in many cities in the Global South, have difficulties in mitigating pandemic impacts and further exacerbate the long-term impact of the lack of access to essential services. Lastly, the origins and cross-border spread of the virus resulted in greater discrimination and stigmatization of foreigners, especially migrants, which may increase pandemic sensitivity factors through shocks to social stability and integration [37].

Economic sensitivity indicators are the most salient determinants of long-term impact, particularly the threat of economic recession. For instance, migrant workers are initially vulnerable to a period of recession. In European countries with rising unemployment rates, opportunities for international migrants are decreasing accordingly [38]. It can also be expected that specialized economic sectors are especially vulnerable to pandemic shocks—tourism is a case in point. In Poland, for example, the cities relying on tourism, mining and manufacturing were the most affected by the pandemic, and the recovery of these industries is more challenging ([39]. Furthermore, small- and medium-sized firms have greater difficulty in absorb the consequences of prolonged restrictions and are more likely to file for bankruptcy.

Transport infrastructure and service accessibility are considered critical factors for the spread of the virus. Early in the outbreak in Italy, the density of trips was strongly related to the infectious cases 21 days later [31]. After a number of new "waves" of the virus, a significant decrease in reported cases has consistently followed after travel restrictions were set. Other papers focused on the resilience of various transport modes. There was a substantial relationship between flights and railway services from Wuhan and new cases in the destination [40]. For other transport modes, Teixeira and Lopes [41] found that in New York the cycling and walking network showed a lower decrease in users than the subway system, along with a modal shift from subway users to bike-sharers. Therefore, motorized transportation modes seem more sensitive to the pandemic, whereas individual, open-air modes are perceived as safer. A final aspect of infrastructural sensitivity is accessibility to health and other daily services, a critical issue whose imbalances became apparent during the pandemic. The areas with low accessibility tend to be living places for poor and marginalized groups, whose higher vulnerability is exacerbated by social sensitivity issues.

Finally, there has been little quantitative evaluation of green and open spaces in relation to the spread of COVID-19, but arguments exist that cities need to increase public spaces for physical distancing and mental health. Providing ample open space can accommodate

residents' recreation demands to meet safely outdoors [42]. The reconfiguration of green space can also improve the urban greenery system, which may contribute to healthier lifestyles and eventually greater resilience against other viruses.

### 2.3.3. Adaptive Capacity

There are many ways to structure the indicators of adaptive capacity [30]. Since the relations between some localized indicators and the metropolitan and urban scales are weak (e.g., investments made only for specific groups only play a limited role in the interconnected urban system) or unstable (e.g., testing rates in different municipalities change continuously), this framework focuses on generic capacity indicators, divided into awareness and ability, which are valid for most populations, areas and institutions for a long time.

Risk awareness is predominantly related to societal maturity, regarding the ability to understand, access and communicate information about the pandemic. For instance, people with poor risk perception and a low level of trust in government have been identified as a high-risk population [43]. This population may often overlap with another vulnerable group, namely, people with low educational levels, who suffer disproportionately from the disease and the financial troubles related to COVID-19 [44], making it even harder to ensure adaptive capacity. The ability to respond to the pandemic is another factor of adaptive capacity. Several authors indicate that adequate investment in primary healthcare systems benefits the effective response to the pandemic, including hospital beds, insurance and social health conditions [43]. In addition, using smart technology is evidenced as a new means to adapt to major social and economic issues through teleworking, online commerce and education and telemedicine [45]. The same authors raised concerns about accessibility and affordability, which require more attention to be given to the the digital divide. Finally, for urban governance, the conflicts between different actors and levels of governance have been exposed by COVID-19. Fragmented governance and inefficient use of limited resources are blamed for the poor management of the spread in some states in the USA [19].

## 3. Applying the Framework in a Pandemic Situation

### 3.1. The Metropolitan Region of Amsterdam (MRA)

Periodic outbreaks characterize pandemics, and urban areas are regularly exposed to crises. Therefore, assessments of pandemic vulnerability are necessary to highlight vulnerable spaces and sectors and propose appropriate strategies for improvement. The Metropolitan Region of Amsterdam (MRA) is the capital region of the Netherlands. It serves as a globally connected region containing compact cities, extensive railway systems and a vast network of highways. The most important economic sectors are financial and professional services, and over the last few decades, the city has developed more international services, such as tourism and hospitality. As in most metropolitan regions, various influences make the MRA particularly vulnerable to COVID-19. By February 2022, the health region of Amsterdam–Amstelland had accumulated over 290,000 cases per million people, above all the other regions of the Netherlands [46]. Therefore, the framework of pandemic vulnerability was tested using this region.

The theoretical framework measures pandemic vulnerability by three components: exposure, sensitivity and adaptive capacity, whose indicators are measuring tools that could indicate how to relieve the impacts of these aspects. For better quantification of these impacts, we started by translating each indicator into representative data taken from various official sources. Table 2 presents this underlying data and its reference metrics. Due to data limitations, the delivery system indicator was excluded from the analysis, and should be taken into when research is available on changes in supply chains during the pandemic. Afterwards, the results of each indicator were spatialized and calculated on a standardized grid with 10 m × 10 m resolution, allowing us to visualize the combined assessment of urban pandemic vulnerability throughout the MRA.

**Table 2.** Measurement of pandemic vulnerability indicators in the context of the MRA.

| INDICATOR | CONCRETE DATA | SOURCE | DATE |
|---|---|---|---|
| **EXPOSURE** | | | |
| Air pollution | NO$_2$ concentration | PDOK | October 2020 |
| Air dryness | Relative humidity | PDOK | October 2020 |
| Import/export nodes | Distance to port and airport | Open Street Map | December 2020 |
| Daily passengers | Heatmap of rail/air passengers | NS | November 2020 |
| delivery system/supply chains | - | | - |
| International corporations | Number of corporations | Open Street Map | November 2020 |
| **SENSITIVITY** | | | |
| Housing conditions | Housing quality | CBS | September 2021 |
| Residential density | Density of inhabitants | CBS | July 2020 |
| Traffic density | Share of daily commuters | Gemeente Amsterdam | December 2019 |
| Hospital accessibility | Distance to hospitals and clinics | Open Street Map | November 2020 |
| Amenity accessibility | Distance to markets and city centers | Open Street Map | November 2020 |
| Open spaces | Distance to parks and playgrounds | Open Street Map | November 2020 |
| Green spaces | Distance to gardens, woods and water | Open Street Map | November 2020 |
| Age distribution | Share of over-70 s | CBS | June 2020 |
| Immigrants | Heatmap of non-European migrants | CBS | March 2020 |
| Social cohesion | Score of social cohesion | CBS | July 2020 |
| Household income | Average household income | CBS | July 2020 |
| Migrant workers | Number of foreign workers | CBS | April 2020 |
| Motorized dependency | Share of bicycle users | Gemeente Amsterdam | October 2019 |
| Tourism | Number of tourists per year | CBS | December 2019 |
| Manufacturing | Number of manufacturing firms/industries | Open Street Map | November 2020 |
| Small businesses | Number of pubs and restaurants | Open Street Map | November 2020 |
| **ADAPTIVE CAPACITY** | | | |
| Hospital beds | Hospital beds per 100,000 inhabitants | CBS | May 2020 |
| Education | Share of residents with tertiary education | CBS | April 2020 |
| Overall health | Mortality rate | CBS | July 2020 |
| Access to digital technology | Density of wifi coverage | CBS | July 2020 |
| Government trust | Trust in Parliament and Government | Gemeente Amsterdam | December 2020 |
| Institutional efficiency | Government activities | Gemeente Amsterdam | December 2020 |
| Institutional integration Risk perception | European Quality of Government Index (EQI) | QoG [47] | 2021 |

Since the indicators refer to very different aspects, their collection and measurement methods—distance or density metrics, survey results, absolute or relative quantities, etc.—had to be adjusted accordingly. Subsequently, the resulting maps of vulnerability could be calculated through a raster analysis of the different indicators. First, each layer of indicators was mapped and clipped or expanded onto the extent of the MRA as a raster layer. Then, the values were normalized to ensure meaningful comparisons and calculations among the indicators. Finally, the following formulas were used in the raster calculator through QGIS to create the vulnerability assessment maps:

**Exposure Index** = (air pollution − air dryness + import/export nodes + daily passengers + international corporations)/5

**Sensitivity Index** = (social sensitivity + economic sensitivity + infrastructural sensitivity + biophysical sensitivity)/4

- *Social sensitivity index = (residential density + age distribution + immigrants − housing conditions + social cohesion)/5*
- *Economic sensitivity index = (−household income + migrant workers + tourism + manufacturing + small businesses)/5*

- *Infrastructural sensitivity index = (traffic density − hospital accessibility − amenity accessibility − bicycle use)/4*
- *Biophysical sensitivity index = (−green space accessibility − open space accessibility)/2*

**Adaptive Capacity Index** = (−education − government trust − hospital beds + mortality − access to digital technology − government efficiency + EQI)/7

### 3.2. Mapping the Results

After adding geographical context, such as built areas, water and green space, Figure 4 shows the final map of exposure. Amsterdam is a city with a generally high pandemic exposure index, especially Amsterdam Centraal, Westpoort and Schiphol Airport (respectively, a railway station, port and airport). These infrastructural nodes are indeed the main entry points of the virus. Measures such as increased testing frequency and stricter lockdowns in these areas could decrease exposure. Compared with Amsterdam, other cities in the MRA, such as Lelystad, were less exposed to COVID-19. However, these cities tended to be affected as well by the resulting spatial and economic problems, as the next sensitivity map shows.

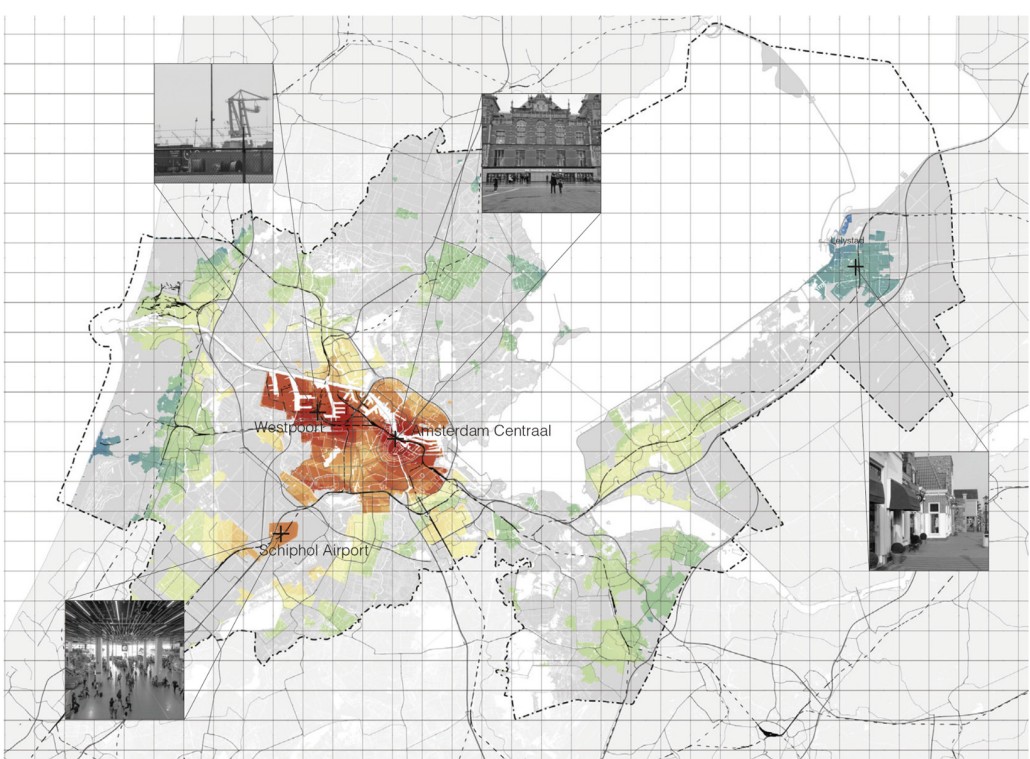

**Figure 4.** Index of exposure in the MRA. Made by the authors.

The sensitivity map in Figure 5 shows that the most sensitive area is Westpoort, the Amsterdam port, which is also a center of manufacturing activities. It has low accessibility to public facilities. Despite its high exposure, central Amsterdam has less sensitivity. Several peripheral neighborhoods are also sensitive. For example, Geuzenveld in Amsterdam Nieuw-West is home to vulnerable migrant communities, who tend to have low incomes and high dependency on public transportation for commuting. Wealthier residential areas, such as Badhoevedorp, Volendam and Nieuw-Loosdrecht, tend to have lower sensitivity indexes. Some people can afford moving from urban centers to suburbs such as these, and often second homes here can be a way to practice remote working, keeping distance from health risks. These areas offer better environments with more open and green space, which also helps to lower their sensitivity indexes.

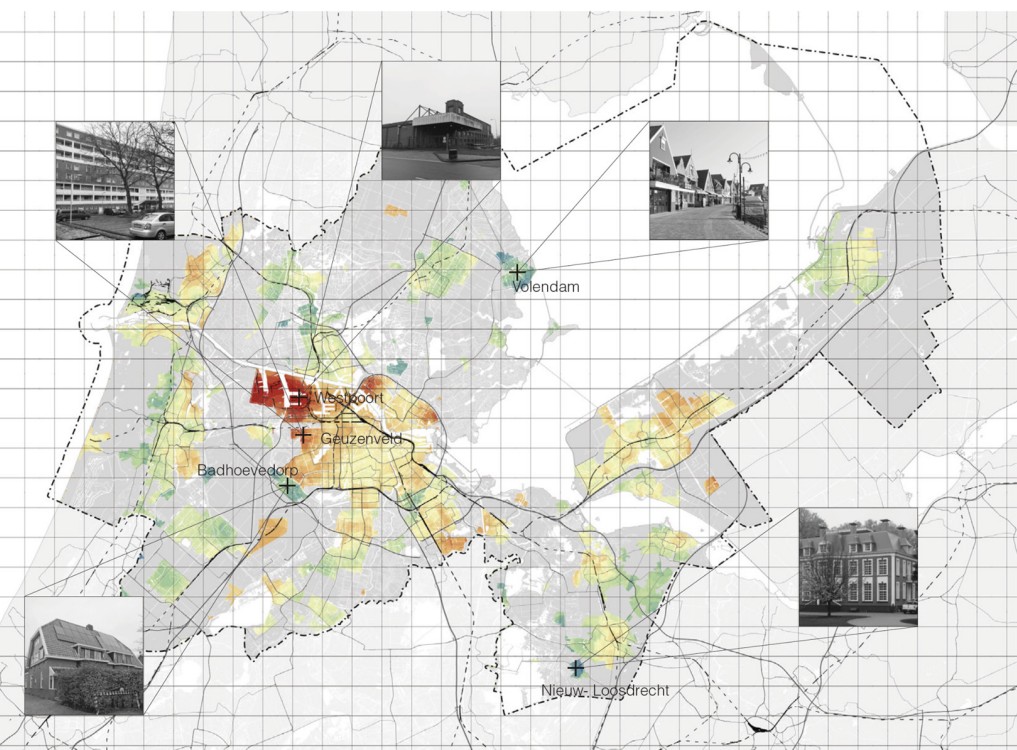

**Figure 5.** Index of sensitivity in the MRA. Made by the authors.

Figure 6 shows the estimated adaptive capacity across the MRA. Unlike the maps of exposure and sensitivity, central Amsterdam shows a high adaptive capacity, probably due to its efficient integration of institutional policies and high allocation of resources to a relatively small spatial setting during the pandemic. Education and health levels of the residents also tend to be high, indicating a clearer understanding and attention to the pandemic crisis, and adequate preparations, such as complying with policies and maintaining social distance. Smaller municipalities such as Almere showed less adaptive capacity, and one of the main reasons could be the lower level of government trust. In 2020, only 40% of residents in Almere believed that the government could control the pandemic in a short time, whereas in Amsterdam 60% of people had confidence in the national government [48]. Slow or unclear institutional responses can seriously affect the timeliness of the outbreak, and the loss of trust among citizens makes it harder to implement measures in these cities.

In conclusion, different areas of the MRA have different aspects of pandemic vulnerability. Therefore, instead of calculating an integrated vulnerability ranking, we present eight vulnerability types, based on different combinations of exposure, sensitivity and adaptive capacity (Figure 7). Each type is visualized as a triangle, whose vertices signal these three components. Since the metrics of the raster maps above are based on scores ranging from 0 to 1, we define high vulnerability as a score higher than 0.5. For the purposes of representation in the triangles, the values are simplified as markers of high value (+) and low value (−) to facilitate practical understanding and quick comparability. The triangles are constructed with higher indexes towards the outer boundaries, except adaptive capacity, whose higher index is negative for vulnerability. Therefore, the larger the triangle's area for a city, the greater the city's pandemic vulnerability. The most vulnerable variant is Type 1, having high exposure, high sensitivity and low adaptive capacity; Type 8 is the least vulnerable, having low exposure, low sensitivity and high adaptive capacity. The eight types are indications of trajectories emerging from the datasets, allowing users to focus on the most vital issues and propose strategies. The simplification conducted here has obvious limitations, such as categorizing an area of the city with little attention to interactions with

surrounding areas and providing an overall indication that neglects extreme values of individual indicators which might have a large impact on the results. However, the value of this exercise as an aid for comparison and decision making, hence its importance as a tool to apply during outbreaks where efficient decisions—even if using plausible but imperfect information—are key.

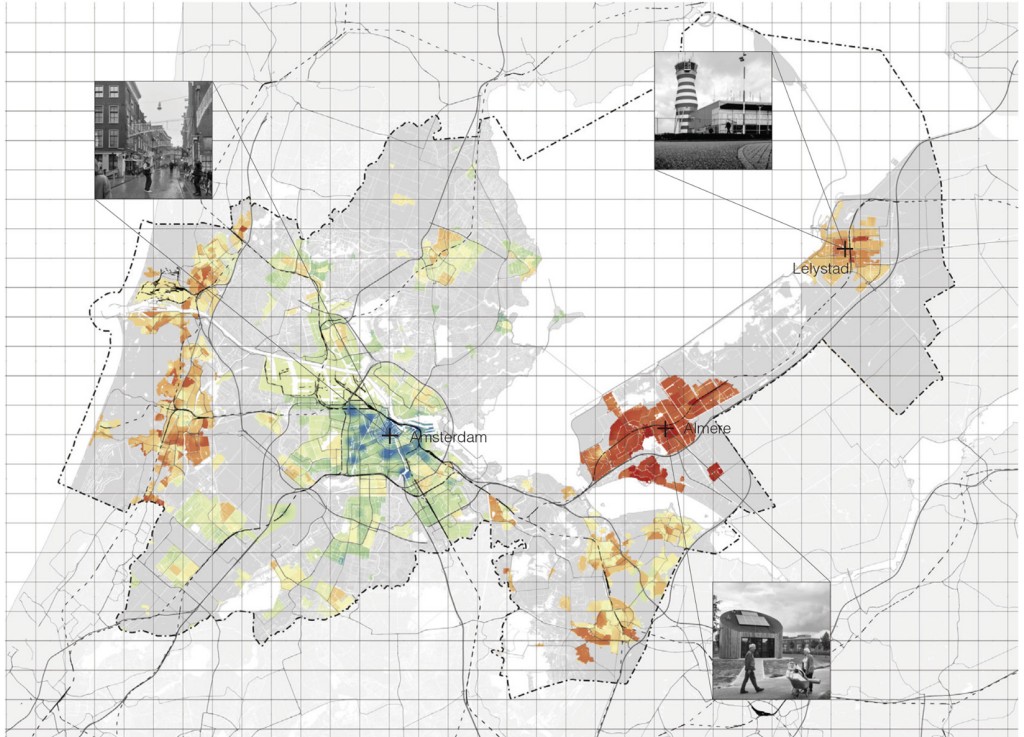

**Figure 6.** Index of adaptive capacity in the MRA. Made by authors.

The final typological map of pandemic vulnerability was calculated based on the results of the three components. First, the raster layers were imported into a GIS file. Then, the values of exposure and sensitivity were transformed into two categories, high value (more than 0.5) and low value (less than 0.5), represented by 1 and 0, respectively. In the case of adaptive capacity, "1" means low value (less than 0.5) and "0" means high value (more than 0.5). Finally, the typological mapping could be calculated as XYZ values through the following formula:

**Typology** = (types of exposure × 100) + (types of sensitivity × 10) + types of adaptive capacity

Therefore, the eight types can be distinguished by raster value. For example, "111" means high exposure, high sensitivity and low adaptive capacity, corresponding to Type 1. Figure 8 shows the typological mapping of pandemic vulnerability in the MRA. The typological distribution shows clear spatial patterns. First, high exposure types are concentrated in the city of Amsterdam. Being the capital of the Netherlands and its most internationally connected global city, Amsterdam was more exposed to the pandemic and the following series of crises. Second, highly sensitive areas are more scattered around the periphery of cities, such as Westpoort and Nieuw-West in Amsterdam, Almere-Buiten and Heemskerk. This reflects that peri-urban areas are often hotspots of sensitive populations, where an unstable economy and inadequate infrastructure can enhance their vulnerability to the pandemic. Thirdly, secondary cities have lower adaptive capacities than Amsterdam. Institutions in these smaller cities may be operating with fewer resources than those in the capital, and some amenities (health, technology) may be comparatively underrepresented. By being less exposed and eventually less sensitive, these cities can suffer less at the

beginning of the outbreak, but due to low adaptive capacity—meaning poor recovery ability between pandemic waves—they may get seriously affected by subsequent waves. This may help explain some worldwide data that show such impact shifts from large to small cities in the course of the pandemic [2].

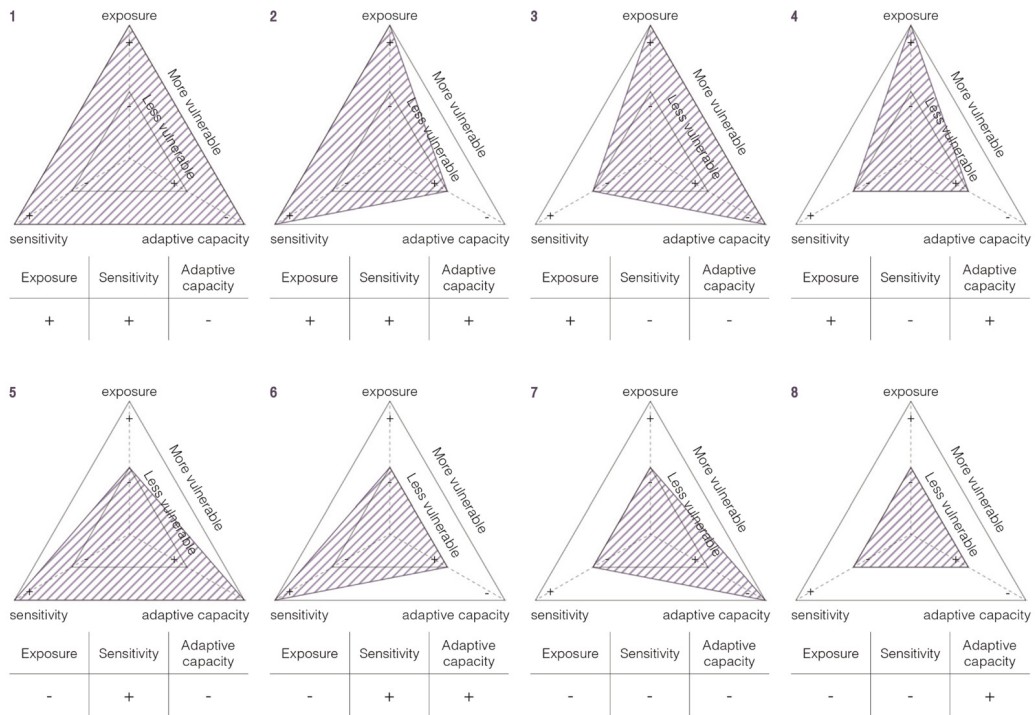

**Figure 7.** Vulnerability types according to exposure, sensitivity and adaptive capacity. High vulnerability scores (above 0.5) are marked with a "+"; low scores (below 0.5) are indicated with a "−".

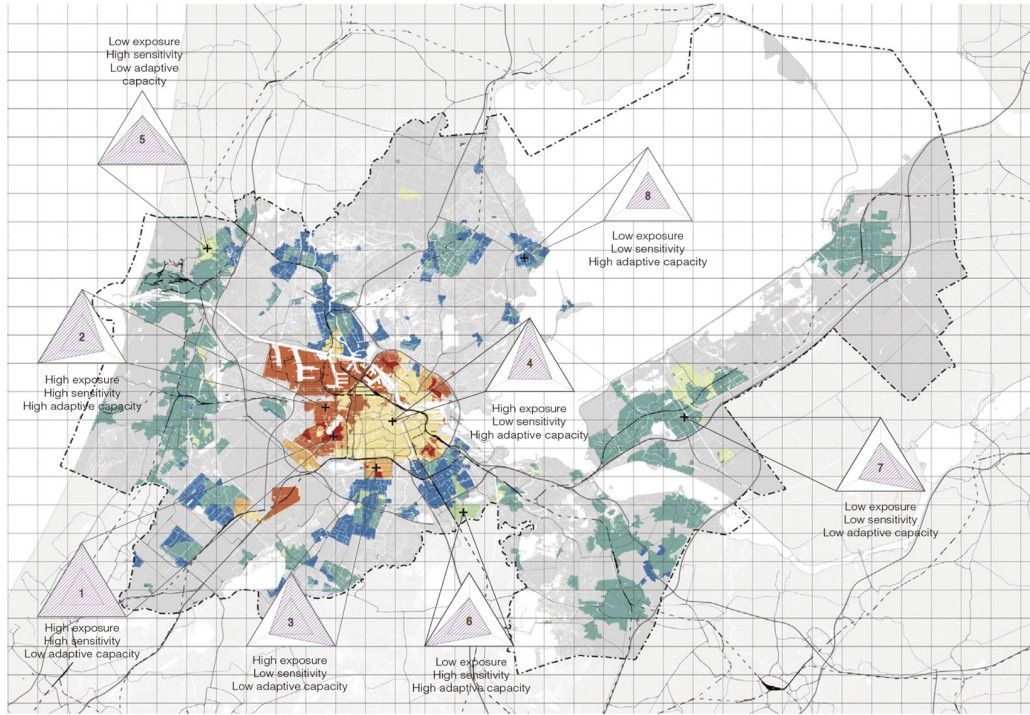

**Figure 8.** Typological mapping of pandemic vulnerability in the MRA.

### 3.3. Identifying Vulnerable Areas

Following our framework, policy responses to the pandemic can be divided into three aspects: decreasing exposure, decreasing sensitivity and increasing adaptive capacity. High exposure and high sensitivity are more difficult and more expensive to combat, and take a longer time to repair, so these were arguably the most overlooked aspects of vulnerability in the MRA response, especially in the first year of the pandemic. Therefore, to identify the most vulnerable areas of Amsterdam, we focused on high exposure and sensitivity areas, giving somewhat lower importance to adaptive capacity. These correspond to areas in the map marked in red and dark orange (Type 1 and Type 2) and include Westpoort (the port area), Schiphol Airport and the district of Amsterdam Nieuw-West. The first two are quite exposed nodes of flows of people, goods, etc., but they are not dominated by residential use; therefore, they can be better protected by measures such as remote work and lockdowns. This is not the case for Nieuw-West, which should be, according to the proposed framework, the most vulnerable area in the MRA, due to potential impacts on social groups, urban spaces and local institutions. Nieuw-West is a densely built district of collective housing, with a lot of greenery between buildings, following the garden city precepts popular in the 1950s and 1960s [49]. The building density, air quality, type of amenities and population profile are some of the indicators that make it both exposed and sensitive to COVID-19 (Figure 9).

In order to test the accuracy of the proposed framework, we checked its predictions against the reality of infection indicators. Additionally, indeed, at the time that the comparison was made (December 2020), the number of infections per capita in Nieuw-West was significantly higher than in other districts. Figure 10 shows the weekly shares of positive-tested people from September 2020 to January 2021 in city districts. The number of hospital admissions was also relatively high, all of which suggest that the spatial analysis, based only on the indicators explained above, was able to pinpoint vulnerability differences quite accurately. Meanwhile, the conditions have changed, and as of November 2021, Nieuw-West no longer has the highest share of positive results—the top spot belongs now to Amsterdam Noord [46]. That district is, according to Figure 8, another hotspot of Type 1/Type 2 vulnerability marked in the map, suggesting the potential of the framework to detect vulnerability indifferent contexts (namely, pre- and post-vaccination programs). Interestingly, Nieuw-West is still problematic in two key indicators—it is the district where fewer people let themselves be tested, and it has the lowest rate of vaccination [46]. Due to distrust of government and skepticism of vaccine information, the willingness of residents to be vaccinated is low [50].

Figure 9 compares the indicators of pandemic vulnerability in Nieuw-West against the Amsterdam average, highlighting the factors that need policy responses. Nieuw-West is exposed through polluted air and import/export nodes. It lies to the south of the port, an important source of pollution in the region, and partly complements its economic activity. The exposure of the port area may therefore put Nieuw-West in greater danger, suggesting the hypothesis of spatial spill-overs of vulnerability hotspots to surrounding areas. Nieuw-West is also sensitive in many aspects of society, economy, infrastructure and biophysics. With over 150,000 inhabitants, the average residential density is higher than Amsterdam, and the district is home to many low-income immigrants: only one-third of the inhabitants are Dutch, and 46% of the population are non-Western. The infrastructural quality is also well below the average of Amsterdam. High traffic density and fewer amenities help make it the district with the lowest share of cycling [51]. Lastly, although Amsterdam overall has higher adaptive capacity than other cities in the MRA, there are some aspects that need to be improved in Nieuw-West. The general level of education and health of the residents is relatively poor. More residents have health problems, such as obesity and other disorders [52], than in other districts. It is known that the coronavirus can be dangerous for obese patients. Additionally, disorders involving physical limitations require more help from family and neighbors, affecting contact restrictions and social distancing.

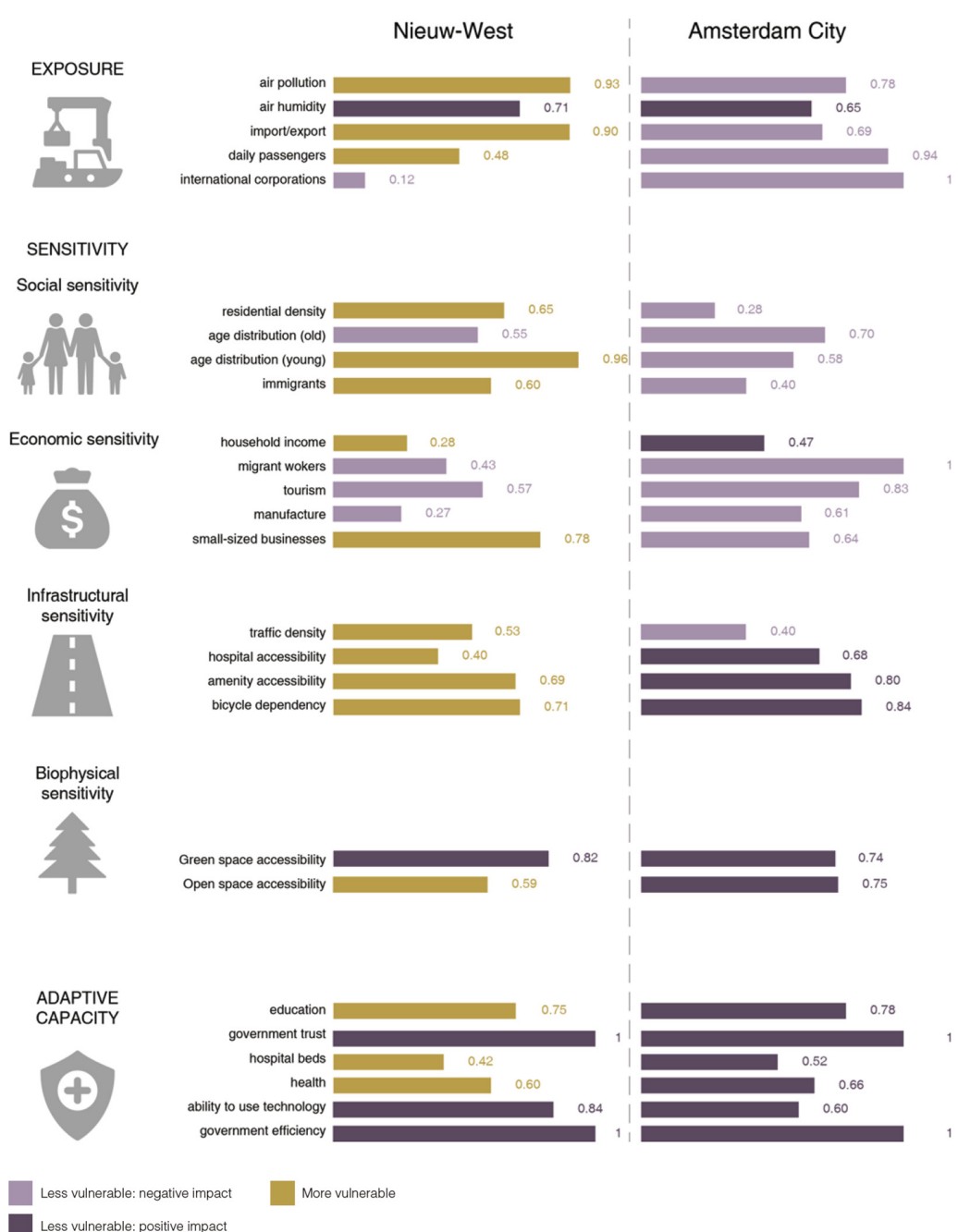

**Figure 9.** Comparing vulnerability scores between Nieuw-West and Amsterdam. Illustrated by the authors.

The accurate detection of vulnerable areas in Amsterdam and the ability to trace back the results of the raster maps to the scores in individual indicators make this model an important tool for policymakers to differentiate districts and neighborhoods; clarify their problems according to the different combinations of exposure, sensitivity and adaptive capacity; identify the policy areas in need of more urgent intervention; and consider appropriate responses.

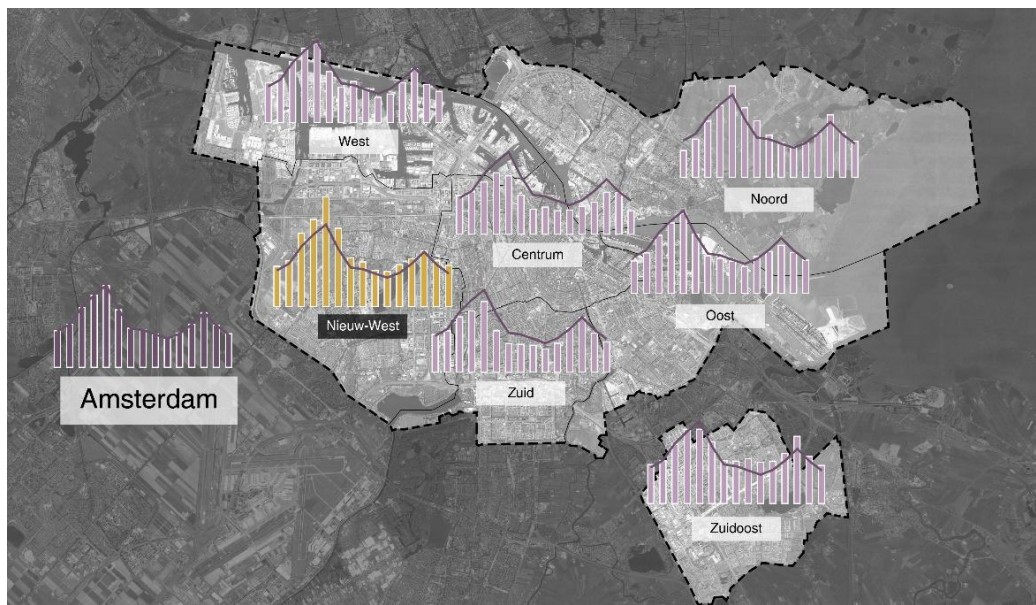

**Figure 10.** Positive cases per districts per 10,000 inhabitants from week 39 (21–27 September 2020) to week 53 (28 December 2020–3 January 2021). Made by the authors; data source: GGD Amsterdam, 24 January 2021.

## 4. Discussion and Conclusions

Metropolitan areas are centers of population and engines of global development and are also on the frontline both as sufferers from, and responders to, the COVID-19 pandemic. Rather than calmly reflecting on slow trends, the academic community is dealing in real-time with the constantly changing urban impacts of COVID-19. A definition and operational framework of urban pandemic vulnerability is needed, since infectious diseases have become a considerable threat both to the interconnected global system and the livelihoods of local communities. Therefore, this paper extends the existing conceptualizations of climate and social vulnerability into a new framework of pandemic vulnerability. Unlike other proposals, the study incorporates both population-level and spatial/environmental indicators, making it more sensitive to urban contexts and arguably more relevant for urban planning. Underpinned by well-supported literature on both climate and social vulnerability, which we found to be partly related to pandemic vulnerability, the framework is based on three components: exposure, sensitivity and adaptive capacity. Critical indicators and respective metrics were identified for each component; we selected those that may enable useful local evaluations and inform preparations for future crises. The framework was then applied to the Metropolitan Region of Amsterdam (MRA). The indicator scores were used for GIS-assisted raster calculations, resulting in detailed maps of exposure, sensitivity and adaptive capacity. Since these components represent different challenges and demand different planning and policy responses, eight typologies were built, based on the combinations of the components, rather than a single vulnerability map, which could make the component scores balance out each other and obscure underlying vulnerabilities. Considering the resulting typologies and their spatial distributions, Amsterdam Nieuw-West was identified as the most vulnerable area in the MRA between March 2020 and January 2021 (pre-vaccination), most of all in terms of exposure and sensitivity, which are arguably the hardest challenges for policy-making.

Despite our best hopes, COVID-19 still cannot be considered a thing of the past, and its potential impact on the world will last. This being a topic changing in real-time, the research on the vulnerability and resilience of cities must be exploratory and constantly attentive to new reviews and observations. Therefore, the framework of urban pandemic vulnerability should be adjusted in face of the latest research, as its components can be different depending on specific spatial contexts, time periods or sociopolitical environments.

Besides, as research on COVID-19 advances, scholars will learn more about the influences of different factors on urban vulnerability. For example, a paper published in April 2021 suggested that climate change may have a direct effect on the spread of coronavirus [53]. Factors related to climate or temperature were not considered in this study, so further exploration is needed to modify the framework. As new waves of the pandemic appear, several new findings will emerge and help fine-tune the model. To support its present reliability, transferability to other metropolitan regions in different countries should be tested. We expect that it is suitable for broader application, as the indicators summarize literature findings from various countries and cities. However, since we simplified our model to enable understanding and analysis, some qualitative dimensions are likely to be neglected. Therefore, evidence from qualitative and quantitative analyses in more metropolitan areas is necessary.

According to this study, it is possible to decrease exposure, decrease sensitivity and increase adaptive capacity, either through measures that help change the scores of individual indicators or by changing the thresholds. Therefore, appropriate design and planning strategies can be developed using those two strategies. On one hand, vulnerability can be reduced by engaging with the indicators, for instance, by improving the ability to absorb impacts without crossing vulnerability thresholds, and supporting quicker recovery from indicator peaks. For example, the indicator of residential density can be changed in a short time by creating more temporary housing during the pandemic. Such measures mitigate specific impacts of the pandemic on specific areas and populations, and emphasize quick responses to ongoing situations. On the other hand, changing the thresholds refers to the ability to change the functioning of the system and move towards a new desirable trajectory of stability. In this way, the threshold of residential density can be changed through a gradual improvement to resilience—for instance, by promoting mixed housing districts. Such measures can decrease the general urban vulnerability in the longer term, giving priority to future generations. Due to its choice of indicators and their respective spatialization, this framework can highlight the vulnerability problems of urban areas in a way that inspires and supports both short and long-term measures.

To conclude, an urban pandemic vulnerability framework can be a planning tool to help urban actors to take measures to decrease vulnerability. This claim is since the model (1) includes spatial and environmental factors which are embedded in urban spaces alongside socio-demographic indicators; (2) is able to detect vulnerable areas at a relatively small scale and with some degree of accuracy supported by real-world evidence; (3) can qualify the type of vulnerability as a combination of exposure, sensitivity and adaptive capacity; and (4) is able to trace back the exact indicators contributing to that particular vulnerability type in order to guide policy responses. These are, in our view, the strong points of this framework, which is necessarily exploratory and subject to continued fine-tuning.

**Author Contributions:** Conceptualization: Y.W.; Methodology: Y.W., R.V.C. and C.F.; Investigation: Y.W.; Writing—Original Draft Preparation: Y.W., R.V.C. and C.F.; Writing—Review and Editing: Y.W., R.V.C. and C.F.; Visualization: Y.W.; Supervision: R.V.C. and C.F. All authors have read and agreed to the published version of the manuscript.

**Funding:** Yaqi Wang and Rodrigo Cardoso received no external funding. The involvement of Claudiu Forgaci in this research was made possible thanks to the funding of the 4TU.HTSF DeSIRE program of the four universities of technology in The Netherlands.

**Institutional Review Board Statement:** Not applicable.

**Informed Consent Statement:** Not applicable.

**Data Availability Statement:** Please see original data sources in Table 2.

**Conflicts of Interest:** The authors declare no conflict of interest.

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
