# Peer review of "Urban Pandemic Vulnerability and COVID-19: A New Framework to Assess the Impacts of Global Pandemics in the Metropolitan Region of Amsterdam"

_sustainability, doi:10.3390/su14074284_

Round 1

Reviewer 1 Report

I find this manuscript highly interesting and informative. But I cannot recommend publication at its current status. Please take the following comments into consideration. 1. I question whether "urban pandemic vulnerability" can be defined by simply combining some aspects of social and climate vulnerabilities. 2. I also question whether the three components of vulnerability (exposure, sensitivity, adaptive capacity) can still be applied to this new "urban pandemic vulnerability." How can the authors be so sure about this? Some of the three components may not work for this new type of vulnerability. Or, new additional components may exist. 3. I am not convinced why the Metropolitan region of Amsterdam was selected for application. It is also highly unclear whether the authors are considering the MRA as a single region or looking into different cities within the region in a comparative way. 4. Relying on quantitative indicators to understand each component may not be comprehensive or misleading. There are many aspects within each component that cannot be quantified and have to be understood in a qualitative way. 5. The vulnerability types (Fig 7) seem to be oversimplifying everything. Can't there be ++, +++, or half a +? This simplified metric (+ or -) does not sound scientific at all. 6. I would be more interested in knowing characteristics of each city as well as what factors among them contribute to different outcomes with regard to their "urban pandemic vulnerability". Don't their demographic characteristics, economic bases, urban form, or connectivity matter, as previous literature has identified, matter? 7. The authors do not prove in any way that adopting this new framework is a better choice than other existing frameworks. What are this new framework's benefits? Aren't there any limitations?

Author Response

Thank you for your comments on our paper. The reviews were useful to make the necessary revisions and we believe that the paper is clearer now. We will now present a detailed response to the individual comments:

  1. I question whether "urban pandemic vulnerability" can be defined by simply combining some aspects of social and climate vulnerabilities.

We wonder about that as well and that is precisely the question we address throughout the paper. In that sense, we do not see this as a critical comment. In section 2.2. we do explain why urban pandemic vulnerability could be defined by combining aspects of social and climate vulnerability frameworks, but of course this is not a settled issue – contributions trying out different ways to address pandemic vulnerability and testing them against each other are certainly necessary. Like most other scholars working on phenomena happening in real-time and where knowledge is incomplete, we are open to contradictory evidence, although the development of our framework and the way it pinpoints the different types of vulnerability in Amsterdam makes us confident that we are providing a useful contribution.

  1. I also question whether the three components of vulnerability (exposure, sensitivity, adaptive capacity) can still be applied to this new "urban pandemic vulnerability." How can the authors be so sure about this? Some of the three components may not work for this new type of vulnerability. Or new additional components may exist.

Again, we are not sure about anything – using these three components is a possible approach that we try to support with literature and examples. And the accurate way in which the framework identifies and differentiates between different types of vulnerability in Amsterdam makes us confident about the approach, but being sure is not an issue here. This is exploratory and future-oriented and future approaches may achieve better results combining other components. However, we do think that it is extremely important to somehow qualify vulnerability – e.g., based on varying intensities and combinations of exposure, sensitivity, and adaptive capacity - because the policy priorities and solutions will differ a lot according to the combination at hand. This qualification ability is somewhat missing in other more generic definitions of vulnerability.

In any case, in response to comments 1 and 2 by this reviewer, we revised the introduction and conclusions to highlight this exploratory/experimental dimension of our contribution and avoid the idea that we are too certain about a single best approach.

  1. I am not convinced why the Metropolitan region of Amsterdam was selected for application. It is also highly unclear whether the authors are considering the MRA as a single region or looking into different cities within the region in a comparative way.

The reasons to select the MRA are duly explained in section 3.1 – the most populated and globally connected urban area in the Netherlands and the fact that it had, at the time of writing, the largest number of cases per million in the country. As the various maps show, the region is treated as a single object where different intensities and combinations of vulnerability emerge in different places. These places can be districts of the core city, or smaller municipalities nearby, such as Almere. We hope the maps are clear enough.

  1. Relying on quantitative indicators to understand each component may not be comprehensive or misleading. There are many aspects within each component that cannot be quantified and have to be understood in a qualitative way.

This is certainly true, but every model-based approach to urban complexity is necessarily a simplification. We revised to conclusions to highlight the fact that some qualitative aspects may have been neglected, but that is the case in all frameworks where choices need to be made. If you look at the list of indicators, you will see that we include social, spatial, and environmental aspects, which is already a step forward from frameworks that tend to look only at social aspects based on individual health, socioeconomic and demographic indicators. That is why we consider our framework useful for urban planning and policymaking. But indeed, full comprehensiveness and absence of simplification is not achievable by this type of approach.

  1. The vulnerability types (Fig 7) seem to be oversimplifying everything. Can't there be ++, +++, or half a +? This simplified metric (+ or -) does not sound scientific at all.

The use of (+) and (-) signs is only for the purposes of representation in the triangles indicating the relative vulnerability combinations in different places. What they mean is now better explained in the text (see section 3.2 and Figure 7). Note that the analysis leading to the identification of the intensity and type of vulnerability is still based on the accurate values of the many indicators. The alleged simplifications only appear at end when we need to communicate the results in an accessible way. (+) and (-) are simple markers often used to assist decision-making processes by non-specialists and we follow that approach here.

  1. I would be more interested in knowing characteristics of each city as well as what factors among them contribute to different outcomes with regard to their "urban pandemic vulnerability". Don't their demographic characteristics, economic bases, urban form, or connectivity matter, as previous literature has identified, matter?

Please note that many of the indicators expressed in Table 2 related precisely to demography, economic bases, urban form and connectivity. See, for instance, age distribution, income, number of small businesses and industries, rail passengers, housing conditions and density etc. So, all the characteristics mentioned are covered. What they don’t refer to is to “each city”, because we disaggregate the analysis at smaller scales, as the maps show, and section 3.2 explains.

  1. The authors do not prove in any way that adopting this new framework is a better choice than other existing frameworks. What are this new framework's benefits? Aren't there any limitations?

Again (see response to the first comments), proving that this is a better choice is not the point of the paper. We want to explore options, compare them with other possibilities, test them against real-world contexts, and, hopefully, see our approach also criticized, adapted or transformed in the future. In any case, the benefits and limitations of the approach are clearly stated in the revised conclusions.

Reviewer 2 Report

I have read the article entitled, “Urban pandemic vulnerability in the context of COVID-19: A new framework to assess the impact of global pandemics in the Metropolitan Region of Amsterdam” submitted to Sustainability for consideration. I found the article highly informative and very intellectually stimulating. There are a few minor English edits needed (e.g., Centre is Center in US), but beyond that I have no other comments. This appears to be a good article.

Author Response

Thank you for the positive comments. We did a general English language editing.

Reviewer 3 Report

The work is very well prepared and developed. It draws on recent literature and refers to the latest urban analysis, research, and theory. It combines knowledge from several sciences, which gives it a multidisciplinary overtone, which is very important nowadays, especially concerning pandemic times. 
The work is very well structured, and it has a clear message. 
I want to make three small suggestions to the authors of the paper, which are worth considering and - perhaps - improving, to increase the article's value. 
The first two suggestions are structural:
1. in the second chapter entitled "A new interpretation of urban vulnerability," I would emphasize, point out - perhaps by changing or detailing subsection 2.2, that we are describing in this chapter the conceptual framework that will be analyzed later in the paper (Figure 2 refers to this). 
2 I would reword or add to the name of section 3, "Applying the framework," e.g., to "Applying the framework in a pandemic situation." - because the point is to indicate in what context we are analyzing the framework

Note 3. relates to the Discussion and is encouraging. I would encourage the paper's authors to be more confident and meaningfully emphasize the value and nature of the article. The merits of the paper should not be leveled in the same sentence with what elements would be worth expanding on. The work is outstanding, and in the discussion, it is worth emphasizing and boasting about that. I encourage authors to be more courageous and to appreciate themselves. 

The above comments are just suggestions that can be implemented. If the authors choose not to do so, it in no way diminishes the value of the work presented. 

Author Response

Thank you for the positive comments. We responded to comments 1 and 2 by detailing the title of sub-section 2.2 to show that the purpose is to construct the final framework and rewording the title of section 3. Regarding comment 3, we are grateful for the praise, but also considering the comments by Reviewer 1 about not showing excessive certainty about our approach, we don’t think that further highlighting the merits of the work is advisable. We are confident about the qualities of our proposal but, as all approaches to phenomena happening in real-time and where knowledge is incomplete, it is necessarily exploratory and incomplete, and is due to be critically addressed, expanded and transformed in the future.

Round 2

Reviewer 1 Report

I thank the authors for revising the manuscript in a way that incorporates at least some of my comments. Things look better know but there is one issue that still requires clarification.

I am still uncertain about figure 7. It still oversimplifies everything. I know the current approach may be somewhat realistic and the authors have to make some subjective judgements; but using such diagrams has clear limitations. I highly recommend the authors to discuss any shortcomings.

Other than this, I am happy to recommend publication. 

Author Response

Thank you for your reaction to the previous revision and new comment. We now include a few sentences before Figure 7 where we mention two main shortcomings of our simplified diagrams and justify its use.